# EZH2 and KDM6B Expressions Are Associated with Specific Epigenetic Signatures during EMT in Non Small Cell Lung Carcinomas

**DOI:** 10.3390/cancers12123649

**Published:** 2020-12-05

**Authors:** Camille Lachat, Diane Bruyère, Amandine Etcheverry, Marc Aubry, Jean Mosser, Walid Warda, Michaël Herfs, Elodie Hendrick, Christophe Ferrand, Christophe Borg, Régis Delage-Mourroux, Jean-Paul Feugeas, Michaël Guittaut, Eric Hervouet, Paul Peixoto

**Affiliations:** 1UMR1098, RIGHT, Université Bourgogne Franche-Comté, INSERM, EFS BFC, F-25000 Besançon, France; camille.lachat@edu.univ-fcomte.fr (C.L.); walid.warda@univ-fcomte.fr (W.W.); christophe.ferrand@univ-fcomte.fr (C.F.); Christophe.BORG@efs.sante.fr (C.B.); regis.delage-mourroux@univ-fcomte.fr (R.D.-M.); jean-paul.feugeas@univ-fcomte.fr (J.-P.F.); michael.guittaut@univ-fcomte.fr (M.G.); 2Laboratory of Experimental Pathology, GIGA-Cancer, University of Liege, 4000 Liege, Belgium; Diane.Bruyere@uliege.be (D.B.); m.herfs@uliege.be (M.H.); E.Hendrick@uliege.be (E.H.); 3Service de Génétique Moléculaire et Génomique, CHU Rennes, F-35033 Rennes, France; amandine.etcheverry@univ-rennes1.fr (A.E.); marc.aubry@univ-rennes1.fr (M.A.); Jean.Mosser@univ-rennes1.fr (J.M.); 4Plate-Forme Génomique Environnementale et Humaine Biosit, Université Rennes1, F-35043 Rennes, France; 5UMR 6290, CNRS, Institut de Génétique et Développement de Rennes (IGDR), F-35043 Rennes, France; 6UMS 3480 Biosit, Faculté de Médecine, Université Rennes1, UEB, F-35043 Rennes, France; 7DImaCell Platform, Université Bourgogne Franche-Comté, F-25000 Besançon, France; 8EPIGENExp (EPIgenetics and GENe EXPression Technical Platform), Université Bourgogne Franche-Comté, F-25000 Besançon, France

**Keywords:** epigenetics, epithelial mesenchymal transition, H3K27me3, EZH2, KDM6B

## Abstract

**Simple Summary:**

Epithelial to Mesenchymal Transition (EMT) has been linked to multiple cancer features including invasion, metastasis, immune escape, and treatment resistance, and has therefore become a promising target mechanism for improving cancer treatment. Since it has also been described that EMT can be reversed by modulating its epigenetic regulation, our study was designed to decipher this epigenetic regulation of EMT, and more particularly its regulation by the H3K27me3 (trimethylation of lysine 27 of H3) mark and the two main enzymes which modulate it: EZH2 and KDM6B. Our results showed that these two enzymes present paradoxical roles during EMT in two cancer cell lines since both overexpression and inhibition of these two proteins led to the induction of EMT. We then identified three new target genes of EZH2 or KDM6B during EMT (*INHBB*, *WNT5B* and *ADAMTS6*) and demonstrated that the modulation of expression of these genes modulated the invasion capacities of the cells. These findings are therefore promising to prevent cancer cells from acquiring an invasive phenotype via the modulation of these new biomarkers of EMT regulated at the epigenetic level.

**Abstract:**

The role of Epigenetics in Epithelial Mesenchymal Transition (EMT) has recently emerged. Two epigenetic enzymes with paradoxical roles have previously been associated to EMT, EZH2 (Enhancer of Zeste 2 Polycomb Repressive Complex 2 (PRC2) Subunit), a lysine methyltranserase able to add the H3K27me3 mark, and the histone demethylase KDM6B (Lysine Demethylase 6B), which can remove the H3K27me3 mark. Nevertheless, it still remains unclear how these enzymes, with apparent opposite activities, could both promote EMT. In this study, we evaluated the function of these two enzymes using an EMT-inducible model, the lung cancer A549 cell line. ChIP-seq coupled with transcriptomic analysis showed that EZH2 and KDM6B were able to target and modulate the expression of different genes during EMT. Based on this analysis, we described INHBB, WTN5B, and ADAMTS6 as new EMT markers regulated by epigenetic modifications and directly implicated in EMT induction.

## 1. Introduction

The crucial role of Epigenetics in Epithelial Mesenchymal Transition (EMT) has recently emerged. Indeed, type III EMT, which has been linked to metastasis and tumor immune escape [1], two hallmarks of cancer, is a reversible process, conferring migration and invasive abilities to cancer cells [2]. EMT is a conserved cellular mechanism which leads to the transition from an epithelial phenotype to a mesenchymal one [3]. The loss of E-Cadherin, together with an increase in Vimentin expression, is a feature of EMT. Different biological or chemical molecules such as TGF-β, hypoxia, cytokines or growth factors are able to induce EMT in specific models [4,5,6,7,8,9,10,11,12]. This process is accompanied by various modifications of the expression of a wide variety of additional epithelial/mesenchymal markers, depending on the EMT model and the inducer used. These modifications are controlled by specific transcriptional factors, so-called EMT-TFs, belonging to the SNAIL, ZEB (Zinc Finger E-Box Binding Homeobox) and TWIST families.

Epigenetic marks, whose roles have been largely demonstrated in cancers, are reversible and control gene expression. It has, for example, been demonstrated that the hypermethylation of the *CDH1* (*Cadherin 1*) promoter is involved in the decrease in expression of the gene during EMT [13]. On the contrary, DNA demethylation has been shown to induce the upregulation of the microRNA, *miR-200c*, and the subsequent inhibition of EMT in clear cell renal cell carcinoma [14]. Similarly, the demethylation of the *PD-L1* (*Programmed Cell Death 1 Ligand 1*) promoter in TGFβ-induced EMT cell models favored the expression of the immune checkpoint inhibitor PD-L1 and led to the inhibition of immune response against cancer cells during EMT [15]. Interestingly, two epigenetic enzymes with paradoxical roles have previously been associated to EMT: (i) the histone methyl transferase (HMT) EZH2 (Enhancer of Zeste 2 Polycomb Repressive Complex 2 (PRC2) Subunit also called ENX-1 or KMT6 for Lysine methyltranserase 6) which is able to add the H3K27me3 mark and, (ii) the histone demethylase (HDM) KDM6B (Lysine Demethylase 6B also called JMJD3) which is able to remove the H3K27me3 mark. In previous studies, it has been shown that the *miR101*-dependent inhibition of expression of EZH2 has been associated with EMT and poor prognosis in oral tongue squamous cell carcinoma [16]. EZH2 expression has also been correlated to invasiveness in laryngeal or breast carcinoma [17,18]. Similarly, KDM6B, although far less studied, has been positively associated with EMT or metastasis in renal cancer or hepatocarcinoma [19,20]. Although these paradoxical roles have been recently discussed [21], it remains unclear how these proteins, with apparent opposite activities, could both promote EMT and/or cancer aggressiveness. To address this question, and in order to better understand the respective roles of EZH2 and KDM6B during EMT, we decided, for the first time, to simultaneously study these proteins in in vivo and in vitro models. 

## 2. Results

### 2.1. Both EZH2 and KDM6B Are Overexpressed during EMT in Patients with NSCLCs 

Since EZH2 and KDM6B have been both associated with EMT and aggressiveness in independent cancer models, we first analyzed their expressions in NSCLCs (Non Small Cell Lung Carcinomas). FFPE (Formalin-*Fixed* Paraffin-Embedded) biopsies were sorted in EMT- and EMT+ groups according to their Vimentin status (Figure 1), as already described in Reference [15]. An increase in both EZH2 and KDM6B stainings was observed in EMT+ tumors compared to EMT-tumors (Figure 1A–C). Increases in the mRNA level of EZH2 and KDM6B are also obtained using an EMT-inducible A549 cell model (using both TGFβ/TNFα) (Appendix A). Moreover, these expressions were also strongly correlated (Figure 1D), independently of the Vimentin status (Appendix A). These results were confirmed using data issued from the retrospective KM-plotter database with JetSet best probes [22]. We showed that lung cancer patients presenting high *EZH2* mRNA levels or high *KDM6B* mRNA levels presented a significantly higher risk of early death (*p* = 3.4 ×10^−5^ and *p* = 4.8 × 10^−5^) compared to patients with lower *EZH2* or *KDM6B* mRNA levels (Figure 1E–G). This risk was further discriminated (*p* = 1.8 × 10^−5^) when patients were compared in regard to the mean expression of *EZH2* and *KDM6B* (high score: high mean *EZH2*/*KDM6B* vs. low score: low mean *EZH2/KDM6B*). 

### 2.2. Effects of Overexpression of EZH2 and KDM6B on EMT Induction

Next, we determined, using an EMT-inducible A549 cell model (using both TGFβ/TNFα), if the overexpression of EZH2 (pMSCV-EZH2-IRES-GFP) or KDM6B (pMSCV-JMJD3 expressing an HA tag) was able to induce EMT (Figure 2). Overexpression of both proteins in stable transfected cells (Figure 2A, Appendix A) did induce morphological changes characterized by a decrease in cell–cell adhesion and an increase in cell length (Figure 2B). Increased cell length was confirmed using phalloidin-rodhamin staining (Figure 2C). EZH2 and KDM6B overexpression also induced an increase in EMT markers at the mRNA level (Figure 2D) characterized by *VIMENTIN, ADAM19* or *MMP2* upregulation and a *CDH1* decrease, as already described using a TGFβ/TNFα treatment. These effects were accentuated in cells overexpressing EZH2. Western-blot experiments show a significant Vimentin increase in cells overexpressing EZH2 and a decrease in E-Cadherin expression in cells overexpressing both EZH2 or KDM6B (Figure 2E). In order to clarify the role of these proteins on cancer phenotypes, we performed migration and invasion assays (Figure 2F-G). As expected, EZH2 and KDM6B overexpression increased these EMT-related phenotypes. To validate the role of EZH2 and KDM6B on EMT induction, we also performed transient transfection of HA-EZH2 and HA-KDM6B (Appendix A) and observed the same increase in EMT markers compared to the levels quantified in the untransfected control cells.

### 2.3. Inhibition of EZH2 and KDM6B Activities Regulated EMT Signaling

We next decided to clarify the function of these enzymes during EMT using specific inhibitors of EZH2 and KDM6B (EZH2i: GSK343 and KDM6Bi: GSKJ4) in EMT cell models. As expected, the inhibitor EZH2i decreased H3K27me3 levels in A549 cells, but according to the data found in the literature [23], we could not observe a significant increase in H3K27me3 signals in these cells when we used the inhibitor KDM6Bi (Figure 3A and Appendix A). Inhibition of KDM6B activity alone, using the specific inhibitor, led to a slight induction of the mesenchymal phenotype (Figure 3B). Indeed, we observed an increase in cell length following an EZH2i or KDM6Bi/TGFβ/TNFα co-treatment compared to the TGFβ/TNFα treatment alone in the A549 cell model (Figure 3C). We next targeted molecular markers to confirm these results. Using RT-qPCR, WB, IF, or flow cytometry, we showed that KDM6Bi induced Vimentin expression (Figure 3D,E and Appendix A). Moreover, a decrease in E-Cadherin and an increase in N-Cadherin stainings were observed after a KDM6Bi treatment in the EMT-induced A549 cell model (Figure 3F,G). Since GSK343 is more specific to EZH2 than to EZH1 (about 60-fold and 1000-fold for others methyltransferases), we next used the chemical inhibitor UNC1999 (EZH1/2i) which targets both enzymes, EZH1 and EZH2. This inhibitor EZH1/2i decreased H3K27me3 levels and promoted Vimentin and N-Cadherin stainings (Appendix A). A phalloidin-rodhamin staining confirmed an increase in cell lengthening following the inhibition of both EZH1 and 2 activities. We finally analyzed whether EZH2i and KDM6Bi could also modulate EMT-related phenotypes, such as cell migration and invasiveness. As expected, KDM6Bi potentiated both cell migration and invasion in EMT-induced A549 cells by TGFβ/TNFα, whereas EZH2i did not (Figure 3H,I). 

*EZH2* and *KDM6B*_CRISPR/Cas9 A549 models also confirmed that the invalidation of both enzymes favored EMT phenotypes (Appendix A). However, in agreement with the results obtained with the chemical inhibitors, CRISPR/Cas9_*KDM6B* cells presented a more detectable mesenchymal phenotype than CRISPR/Cas9_*EZH2* cells, with a higher expression of Vimentin and a significant increase in cell migration ability compared to control cells. 

To confirm that our results obtained using the A549 model induced in EMT using TGFβ/TNFα were not dependent of the cell line, or the inducer, we also used a second model of EMT induction, the MDA-MB-468 cells in which EMT was induced using EGF. In this second model, overexpression of EZH2 or KDM6B was also associated with a higher Vimentin staining. Moreover, EZH2i, and KDM6Bi at a lower level, promoted EMT-related phenotypes in this second model, as well (Appendix A).

### 2.4. The Genes Targeted by EZH2 and KDM6B during EMT Were Different

Since we demonstrated that both enzymes were able to promote EMT, even if they present an apparent opposite activity, we speculated that these surprising results might be linked to the targeting of different genes during EMT. So, to answer this question, we performed ChIP-seq (chromatin immunoprecipitation-sequencing) experiments targeting EZH2, KDM6B, and the H3K27me3 mark in A549 cells treated with TGFβ/TNFα. We then sorted 9493 loci presenting an enrichment of both EZH2 and the H3K27me3 mark and 13,122 loci associated with an enrichment of only KDM6B (Figure 4A). As expected, the number of loci targeted by both EZH2 and KDM6B was very low (<3%), suggesting that the recruitment of these enzymes was indeed exclusive. Overlap and downstream positions of these loci were then crossed with previous transcriptome results obtained in our laboratory [24], and we then identified 169 down-regulated genes during EMT which might be controlled by H3K27me3 methylation in an EZH2 manner, and 566 up-regulated genes during EMT which could be regulated by KDM6B (Figure 4A). A list of the 40 most down-regulated genes presenting an increased enrichment of both EZH2 and the H3K27me3 mark on the gene or the promoter was summarized in Table 1. Similarly, the 40 most up-regulated genes presenting an increased enrichment of KDM6B alone was summarized in Table 2. Amongst these genes, many have been previously associated with EMT features, such as adhesion, migration, invasion, cytoskeleton remodeling, or differentiation. 

Interestingly, our ChIP-seq results revealed enrichment of KDM6B on the *SNAI1*, *ADAM19*, (*ADAM Metallopeptidase Domain 19*), and *MMP9* genes, but not on the *ZEB-1* gene. Moreover, an increase in *SNAI1*, *ADAM19*, and *MMP9* expression was observed when A549 cells were treated with EZH2i or KDM6Bi combined with a TGFβ/TNFα treatment compared to a TGFβ/TNFα treatment alone (Figure 4B). But, similarly to our previous results, the induction of expression was higher when we used KDM6Bi rather than EZH2i. As expected, the expression of the *ZEB-1* gene was not modified by EZH2i or KDM6Bi, the two enzymes being unable to bind to this core promoter. We then confirmed, using single ChIP experiments, that a decrease in the H3K27me3 mark, which was potentiated by EZH2i, was observed on *ADAM19* and *MMP9* promoters in cells treated with TGFβ/TNFα (Figure 4C). 

### 2.5. INHBB, WNT5B and ADAMTS6 Established as New Epigenetic Biomarkers of EMT

In order to identify new target genes of EZH2 and KDM6B during EMT which could be later used as epigenetic biomarkers during EMT, we next confirmed our transcriptome and ChIP-seq results using single H3K27me3-targeted ChIP experiments and RT-qPCR analysis. Our data showed that, in agreement with the data presented in Table 1, *INHBB* (*Inhibin Subunit Beta B*) expression was significantly decreased during EMT and associated with a gain of the H3K27me3 mark on its promoter in EMT-induced A549 cells. On the contrary, in agreement with the data presented in Table 2, *WNT5B (Wnt Family Member 5B)* and *ADAMTS6* (*ADAM Metallopeptidase with Thrombospondin Type 1 Motif 6*) expressions were increased during EMT and associated with a loss of the H3K27me3 mark on their respective promoters (Figure 5A,B). To determine whether the modulation of the levels of H3K27 methylation was inversely correlated to the level of H3K27 acetylation (H3K27ac, a permissive mark [25]) in EZH2/KDM6B-targeted genes and to characterize the timing of these epigenetic modifications during EMT, we performed kinetic ChIP experiments (Figure 5C). The data obtained validated our hypothesis that an increase in the H3K27ac content was correlated with a decrease in the H3K27me3 level, and inversely, on *INHBB*, *ADAMTS6*, or *WNT5B* promoters. Our data, therefore, validated the idea that the H3K27 mark was tightly regulated during EMT and might be a determinant of controlling gene expression. Moreover, these kinetic analyses indicated that the modulation of H3K27 methylation levels was progressive during EMT induction, with the highest effect obtained after 72 h of induction. 

Next, we analyzed the expression of INHBB, ADAMTS6, and WNT5B at the protein level in NSCLCs (Figure 5D,E). The expression of these markers was assessed in EMT- (*n* = 25) and EMT+ (*n* = 24) tumors using IHC. Our in vivo data were in agreement with our in cellulo results since we observed a significant increase in WNT5B and ADAMTS6 expressions in EMT+ vs. EMT-tumors while the expression of INHBB was significantly decreased in EMT+ compared to EMT-tumors. 

We next wondered whether our targets were directly involved in EMT-related phenotypes following the invalidation of *INHBB*, *WNT5B*, and *ADAMTS6* using specific siRNAs (Figure 5F). Interestingly, the invalidation of *INHBB* expression, without the use of an EMT inducer, was sufficient to increase invasion (Figure 5G). As expected, since a TGFβ/TNFα treatment completely abolished *INHBB* expression, no additive effect of the combination of *siINHBB* and TGFβ/TNFα could be observed. On the opposite, the combination of TGFβ/TNFα with a *siWNT5B*, or with a *siADAMTS6*, decreased invasion. These data strongly supported the idea that the invalidation of *INHBB* expression mediated by H3K27 methylation and the promotion of *WNT5B* and *ADAMTS6* expressions mediated by H3K27 demethylation favored invasion during EMT. Altogether, our data demonstrated that *INHBB*, *ADAMTS6*, and *WNT5B* could be considered as three new EMT biomarkers, epigenetically regulated during EMT.

### 2.6. Effect of EZH2 and KDM6B Inhibition on INHBB, WNT5B or ADAMTS6 Expression

To analyze whether the inhibition of EZH2 and KDM6B was able to modulate *INHBB*, *WNT5B*, or *ADAMTS6* expression, we performed RT-qPCR experiments in the presence of the different enzyme inhibitors (Figure 6A). As expected, EZH2i decreased the levels of the H3K27me3 mark on the three promoters (Figure 6B). Neither EZH2i (Figure 6A) nor *siEZH2* (Appendix A) were sufficient to induce *INHBB* expression during EMT, even if H3K27me3 levels were decreased after EZH2i treatment. However, EZH2i was able to induce an increase of *WNT5B* expression in cells co-treated with TGFβ/TNFα (Figure 6A). Similar results were observed when EZH2 expression was silenced (Appendix A). Regarding *ADAMTS6*, unexpected results were observed with EZH2i since we observed a decreased expression, whereas, as expected, *siEZH2* increased its expression in the presence of TGFβ/TNFα. These results indicated that the modulation of these enzymes partially modulated the expression of our targeted genes, probably due to additional post-translational histone marks, such as deacetylation, which could not be reversed by these treatments. Our results (Figure 5C) might also suggest that H3K27me3 is a late mark modified to stabilize the effects on gene expression, rather than to induce the initial modulations.

## 3. Discussion

Both EZH2 and KDM6B, two proteins with opposite catalytic activities, have previously been associated with cancer aggressiveness or EMT. Indeed, a high expression of EZH2 was associated with metastasis and a poor clinical outcome in clear cell renal cell carcinoma, esophagus squamous cell carcinoma, bladder urothelial carcinoma, cutaneous melanoma, hormone-refractory and metastatic prostate cancer, gastric carcinoma, breast carcinoma, lung carcinoma and ovarian carcinoma, as well as in pediatric brain tumors (for a review, see Lachat et al. [21]). However, KDM6B expression has also been correlated with metastasis in clear cell renal cell carcinoma, ovarian cancer, myeloma, hepatocellular carcinoma, malignant pleural mesothelioma, and Hodgkin’s lymphoma or invasive breast tumors. To clarify this apparent paradox or to understand whether these proteins were involved in metastasis in specific models, we decided to analyze the role of both EZH2 and KDM6B in the same EMT cellular model. Indeed, the majority of previous investigations were focused on only one epigenetic enzyme, and our study highlighted the complexity of simultaneously analyzing two epigenetic modulators with opposite functions. Therefore, for the first time, we observed that EZH2 and KDM6B expressions were strongly correlated in NSCLCs and both enzymes correlated with EMT in these tumors (Figure 1) and with poor clinical outcomes. These data strongly supported the idea that both enzymes were involved in EMT and activated pro-EMT signaling pathways and that this induction only required one of these proteins as most people suggested in the literature. Our data also suggested that the expression of these two enzymes was co-regulated in lung cancers, but this last point would need further investigation to be confirmed.

To clarify the roles of EZH2 and KDM6B during EMT, we modulated the expression and/or activity of these enzymes separately. As expected, both overexpression of EZH2 or KDM6B promoted Vimentin expression, a well-known EMT marker (Figure 2). Interestingly, the inhibition of KDM6B activity alone, using a specific inhibitor, was sufficient to promote EMT signaling (Figure 3). Nevertheless, the EZH2 or KDM6B inhibitor potentiated the effect of TGFβ/TNFα to induce EMT, and a more pronounced phenotype was observed when the activity of KDM6B was inhibited compared to the inhibition of EZH2. These data confirmed that the modulation (increase or decrease) of expression of each enzyme promoted EMT. Das and Taube [26] indeed highlighted the difficulty of addressing the precise contribution of these enzymes in a recent review. 

The results of Figure 1 and Appendix A suggest that both enzymes were strongly co-regulated and might be each responsible for the regulation of a specific panel of genes during EMT. Indeed, EZH2 would be involved in the repression of a specific subset of genes, whereas KDM6B would activate different targets. Our data also highlighted that the down- or up-regulation of one enzyme might be compensated by an increase, or inhibition, of the other. For example, it has been shown that PRC2, whose EZH2 is the catalytic member, could regulate different subsets of genes depending on the cell type and could function, at least partially, independently of the EMT-TFs. Indeed, EZH2 did not regulate *SNAI1* or *SNAI2* gene expression in head and neck squamous cell carcinoma cell lines, and the inhibition of its expression repressed the expression of mesenchymal markers and inhibited migration and invasion capacities without modulating EMT-TF levels [27]. Similarly, KDM6B has also been associated with the direct activation of the EMT-TF gene *SNAI1*, or *SNAI2*, in different cancer models (breast cancer cells [28], hepatocarcinoma cells [20], and CRCCs [19]) but KDM6B might also be recruited on specific promoters involved in downstream EMT signaling. Therefore, to identify the promoters which could be targeted by EZH2 or KDM6B, we performed a ChIP-seq analysis in A549 EMT-induced cells (Figure 4). We first confirmed that EZH2 and KDM6B occupied distinct sites in the genome and that most sites targeted by EZH2 also presented the H3K27me3 mark, while KDM6B was mostly associated with genomic sites in which the H3K27me3 mark was absent. Although *CDH1* is a well-known repressed gene by EZH2 [29,30], our study suggested, for the first time, that numerous additional genes were also controlled by EZH2 and KDM6B during EMT. Interestingly, it has been previously reported in liver cancers that EZH2 repressed, in an H3K27me3-dependent manner, the *APC* promoter leading to the activation of the Wnt/β-Catenin pathway [31]. Interestingly, our study confirmed the presence of both EZH2 and the H3K27me3 mark on the *APC* promoter (data not shown) demonstrating the validity of our model and experimental scheme.

Amongst the genes specifically targeted by EZH2 and KDM6B during EMT, we confirmed a decreased expression of *INHBB* and an increased expression of WNT5B and ADAMTS6 (Figure 5). *INHBB* was the gene presenting the most important decrease in expression (FC = 31) associated with concomitant recruitment of EZH2 and the presence of the H3K27me3 mark. Interestingly, this gene was previously identified as a regulator of hormone secretion from the pituitary gland (for a review, see Reference [32]) but, more recently, INHBB expression has been associated with cancer, although its role in EMT remains controversial. Indeed, high Activin B levels, a homodimer of INHBB, have been associated with a higher risk of metastasis in oral squamous cell carcinoma [33] but, INHBB expression was more recently associated with an inhibition of cell migration in nasopharyngeal carcinomas [34]. *ADAMTS6* and *WNT5B*, whose expressions were increased (FC = 60 and 49, respectively) and associated with the presence of KDM6B on their promoters, have been associated with EMT in recent publications, although their roles are still poorly investigated in this process [35,36,37]. ADAMTS6 is one of the 19 members of the metallopeptidase ADAMTS family, but its physiological role is unknown. For example, its expression has been associated with invasion and cancer recurrence in pituitary tumors and poor prognosis (clinical stage, lymph node metastasis and recurrence) in esophageal squamous cell carcinoma [37,38]. Similarly, WNT5B has been very recently identified as a key regulator governing the aggressive basal-like phenotype in breast cancers [39]. Moreover, WNT5B appeared necessary to promote invasion and secondary EMT in a non-canonical WNT pathway manner [35]. We confirmed, by selectively inhibiting the expression of these genes, that the inhibition of *INHBB* promoted cell invasion, whereas the inhibition of *WNT5B* or *ADAMTS6* decreased cell invasion in EMT-induced cells (Figure 5). Moreover, an increased expression of ADAMTS6 and WNT5B and a decreased expression of IHNBB were confirmed in EMT-positive compared to EMT-negative NSCLCs (Figure 5). Although EZH2 or KDM6B were found on the promoters of these genes, the regulation of their expression seemed only partially regulated by the H3K27me3 mark. Indeed, the inhibition of EZH2 or KDM6B partially controlled the expression levels of these new targets. These partial effects might be due to additional epigenetic modifications or the fact that H3K27me3 might only stabilize previous modulations of gene expression rather than to initiate these modulations. To further characterize the list of genes, directly and fully, controlled by EZH2 and KDM6B, new ChIP-seq experiments combined with transcriptomic analysis will be performed following the inhibition of EZH2 or KDM6B. 

## 4. Materials and Methods 

### 4.1. Cell Culture, Transfections, and Inhibitors Used in Cell Culture

The A549 (NSCLC) cell line was obtained from Dr Christophe Borg (INSERM UMR1098, Besançon, France) and the MDA-MB-468 cell line was obtained from Dr Christine Gilles (Laboratory of Tumor and Developmental Biology, Liège, Belgium). A549 cells were grown in DMEM 1 g/L glucose (L0066, Dominique Dutscher, Brumath, France) and MDA-MB-468 were cultured in RPMI 1640 medium. Both media were supplemented with fetal calf serum (respectively 5 and 10%) (S1810, Dominique Dutscher), penicillin- streptomycin (50 U/mL) (L0018, Dominique Dutscher) and amphotericin B (1.25 µg/mL) (P11-001, PAA). Cells were cultured at 37°C in 5 % CO_2_, and routinely used at 70−80% confluence. When indicated, cells were treated for 3 days using 4 ng/mL TGFβ (100-21, Peprotech) and 20 ng/mL TNFα (300-01A, Peprotech, Rocky Hill, NJ, USA) or EGF (AF-100-15, Peprotech) for 2 days. GSK343 (SML0766, Sigma-Aldrich, St. Louis, MI, USA), GSKJ4 (SML0701, Sigma-Aldrich) were used at 10 µM or 5 µM (in A549 cells or MDA-MB-468 cells, respectively), and UNC1999 (#46080, Cell Signaling) was used at 5 µM. Transfections of plasmids were performed using the JetPrime reagent (#114, Polyplus, New York, NY, USA) or with the Cell line Nucleofector™ Kit T (VCA-1002, Lonza, Basel, Switzerland) using the Nucleofector™ 2b Device (AAB-1001, Lonza) following the supplier’s recommendations. The pCMV-HA-EZH2 (#24230), pCMV-HA-KDM6B (#24167), pMSCV-JMJD3 (#21212) and pMSCV-EZH2-PGK-Puro-IRES-GFP (#75125) vectors were obtained from Addgene. A stable wild type JMJD3 cell line was obtained using pMSCV plasmids via transduction of the A549 cells with lentiviral particles as described in Reference [40]. The selection of transduced cells overexpressing EZH2 or KDM6B was performed under a puromycin treatment at 3 µg/mL for 1 week. siRNA *EZH2* (Dharmacon, Catalog ID:L-004218-00-0005), siRNA *KDM6B* (Dharmacon, Catalog ID:L-023013-01-0005) and siRNA *ADAMTS6* (L-005-776-00-005, Dharmacon, Lafayette, CO, USA) were obtained from Dharmacon and siRNA *WNT5B* (5′-CUCCUGGUGGUCAUUAGCUUU-3′, 5′-GCTAAUGACCACCAGGAGUUU-3′, Eurogentec, Madison, WI, USA) and siRNA *INHBB* (sc43861, Thermo Fisher Scientific, Santa Cruz, CA, USA) were transfected using Interferin (409-10, Polyplus) according to the manufacturer’s protocol.

### 4.2. Invasion and Migration Assays

Migration assays were performed using a 2-well culture-insert in 35 mm microdishes (81176, ibidi). A total of 18,000 pretreated cells were plated in the 2 wells, and the inserts were removed 24 h later. Migration was monitored for 8 h. For Invasion assays, 50 µL of ECM Gel from Engelbreth-Holm-Swarm murine sarcoma (diluted 8 fold in free serum medium) (E1270, Sigma-Aldrich) was added in the upper chamber (140629, Thermo Scientific, Waltham, MA, USA), positioned in 24-multiwell dishes, and incubated for 4 h at 37 °C for polymerization. Then, 50,000 pretreated cells were seeded into the upper chamber in presence of the desired treatment, diluted in free serum medium, and 500 µL of complete medium containing the same treatment was added into the lower chamber. The insert was removed 24 h later, rinsed with PBS, and invasive cells present in the well (cells which migrated through the matrigel and attached to the bottom of the well) were fixed with ethanol 100% for 5 min, stained with crystal violet for 10 min, and then counted. 

### 4.3. Quantitative RT-PCR

RNA was isolated from cells using the Tri Reagent (TR118, Molecular Research Center, Cincinnati, OH, USA) as described by the manufacturer. Reverse transcription was performed using 12 U of M-MLV (M-1302, Sigma-Aldrich) reverse transcriptase, 0.25 µM oligodT (Eurogentec, Liège, Belgium), 1.25 µM random hexamers (C118A, Promega), and 1.5 μg of total RNA according to the manufacturer’s instructions (Sigma-Aldrich). Quantitative PCR (qPCR) was performed in duplicate using the Step one plus Real-Time PCR system (Applied Biosystems, Foster City, CA, USA), the TB Green Premix Ex Taq (Tli RNase H Plus) kit (R420L, Takara, Ann Arbor, MI, USA), and specific primers (Eurogentec) according to the manufacturer’s instructions. The primers were designed using the Primer 3 software and are listed in the Appendix A. 

### 4.4. ChIP/ChIP-seq

Washed cells were fixed with 1 % formaldhehyde at RT for 8 min with gentle agitation and then quenched with the addition of 125 mM Glycine for 5 additional minutes. Cells were centrifuged for 5 min at 500 g and washed twice with 1× cold PBS. Chromatin was prepared using the truChIP™ Chromatin Shearing Kit (520154, Covaris, Woburn, MA, USA) according to the manufacturer’s instructions. Nuclei were submitted to 10 min of sonication using the M220 Covaris sonicator. ChIP was performed using the IP-Star Compact Automated System (B03000002, Diagenode, Denville, NJ, USA) with the Auto iDeal ChIP-seq kit for Histones (C01010171, Diagenode) and 1.5 µg of ChIP-grade antibody or IgG (IG07-2, P.A.R.I.S.; C15410206, Diagenode). Protein inhibitor cocktails were added in all buffers. Inputs and ChIP products were purified using the Auto Ipure kit V2 (C03010010, Diagenode) and IP-Star Compact Automated System (Diagenode). For ChIP-seq, libraries were prepared from 5 ng of ChIP DNA and Input DNA with the NEBNext® Ultra™ DNA Library Prep Kit for Illumina (E7370S, New England Biolabs, Ipswich, MA, USA) according to the manufacturer’s instructions. From each library, 50 bp single-reads were sequenced using an Illumina Hiseq 1500 system (Illumina, San Diego, CA, USA). Reads were filtered according to their quality (Q Score ≥ 30), and adapter sequences were removed using Cutadapt [41]. Reads were aligned to the human genome (hg19) using the BWA (version 0.7.10). H3K27me3, EZH2, KDM6B differentially enriched regions (peaks) were identified using SICER-df (g = w = 200 bp). Primers used for classical ChIP were designed using the Primer 3 software (http://primer3.ut.ee/) and listed in the Appendix A. Antibodies used for ChIP are listed in the Appendix A.

### 4.5. Western-Blotting

For total protein extracts, cells were scraped, harvested, and lysed in lysis buffer (10 mM Tris, 1 mM EDTA, 1 mM PMSF, 1 mM NA-Vanadate, 1% DOCA) supplemented with protease inhibitors (104 mM AEBSF, 1.5 mM pepstatin A, 1.4 mM E-64, 4 mM bestatin, 2 mM leupeptin, 80 µM aprotinin) for 30 min on ice and sonicated for 15 s. Protein quantification was performed using the Bradford method, and then proteins (25−40 μg) were separated on TGX acrylamide gels (1610172, Biorad) at 300 V for 30 min using the Protean 3 system and transferred onto Transblot turbo PVDF (1704157, Bio-Rad) membranes for 10−15 min using the Transblot turbo (1704150, Biorad) according to the manufacturer’s recommendations. Membranes were saturated in 0.1% TBS-Tween 20 and 5% milk or BSA (according to antibodies datasheets) for 1 h and then incubated with primary antibodies diluted according to manufacturer’s instructions in 0.1% TBS-Tween 20 and 5% milk or BSA (Appendix A) overnight at 4 °C. Membranes were washed 3 times 10 min with TBS-Tween 20 0.1%, incubated with secondary anti-rabbit HRP conjugate (1/10,000, BI2407, Abliance, Compiègne, France) or anti-mouse HRP conjugate antibody (1/10,000, BI2413C, Abliance) according to manufacturer’s instructions. The membrane was then washed 3 times for 5 min with TBS-Tween 20 0.1% and incubated with the Clarity Western ECL substrate (1705051, Biorad, Hercules, CA, USA) and chemiluminescence was monitored using a Biorad ChemiDoc^TM^XRS+.

### 4.6. Confocal Microscopy

Cells were cultured for 24 h on coverslips and then fixed with cold methanol for 20 min at −20 °C. Blocking was realized with 1% BSA for 1 h at 37 °C. Incubations with primary antibodies were performed overnight at 4 °C, and then cells were rinsed 3 times with 0.1% tween-TBS. Incubations with secondary antibodies goat anti-rabbit and goat anti-mouse AlexaFluor 488 or 555 (1/1000, A11008, A21428, A11001, A21422, Life technologies, Carlsbad, CA, USA) were performed for 1 h at 37 °C and cells were then rinsed 3 times with 0.1% tween-TBS, stained with DAPI (4′,6′-diamidino-2-phénylindole) and mounted using Fluoromount Aqueous Mounting Medium (F4680, Sigma Aldrich). ACTIN was stained with Phalloidin-Rhodamin (P1951, Sigma-Aldrich, Saint-Quentin-Fallavier, France). Images were collected with an Olympus FV1000 or Zeiss LSM800 AiryScan laser scanning confocal microscope with a 63X objective.

### 4.7. Immunohistochemistry 

FFPE biopsies of NSCLCs were collected (in collaboration with the Tissue Biobank of the University of Liege (Liege, Belgium)) and stained with the listed antibodies (Appendix A). The protocol was approved by the Ethics Committee of the University Hospital of Liege and the diagnosis of each case was confirmed by experienced histopathologists. IHC analysis was performed with a standard protocol detailed previously [42]. Samples were classified into two groups: EMT-positive vs. EMT-negative depending on their Vimentin staining. According to Reference [43], IHC were scored by multiplying two values: intensity (0−3) and extent (0−3), leading to a global score ranging between 0 and 9. All immunolabelled tissues were evaluated by two experienced histopathologists. 

### 4.8. Statistics

Mean comparisons were analyzed using a Student *t*-test performed with the GraphPad Prism5 software (San Diego, CA, USA). Correlation indexes were measured using a Pearson test performed with the GraphPad Prism5 software. Significant values were highlighted in bold in each figure. 

## 5. Conclusions

The epigenetic proteins EZH2 and KDM6B, two enzymes controlling the methylation level of the histone H3 at the position K27 (H3K27), are major components of cancer aggressiveness and EMT. Altogether, our results demonstrated, for the first time, that both EZH2 and KDM6B were involved in EMT. Moreover, we showed that the H3K27me3-mediated repression of *INHBB* and the activation of *WNT5B* or *ADAMTS6* expression, linked to the loss of the H3K27me3 mark, could be considered as new biomarkers of EMT in vitro and in vivo, which could be later quantified to better stratify patients. Moreover, the targeting of EZH2 and/or KDM6B might offer new perspectives of therapies to fight metastasis apparition and formation in the future. 

## Figures and Tables

**Figure 1 cancers-12-03649-f001:**
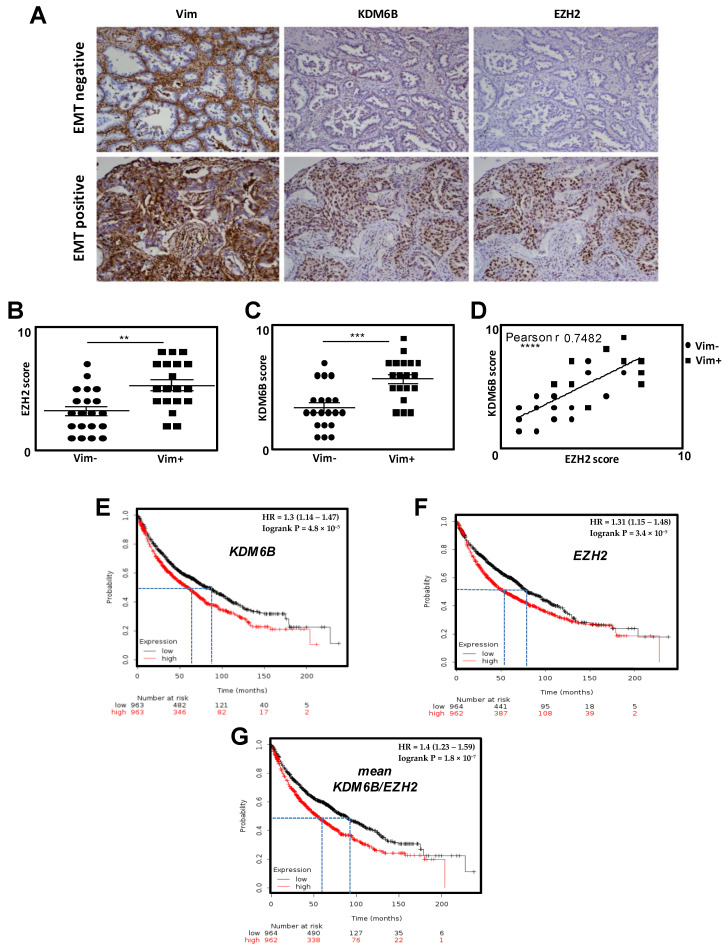
EZH2 and KDM6B expressions are correlated with EMT in NSCLC. (**A**) Representative IHC stainings of Vimentin (Vim, high score: EMT positive/low score: EMT negative), KDM6B, and EZH2 in 30 NSCLC (magnification × 100). (**B**) Quantification of the IHC EZH2 score regarding the EMT status. (**C**) Quantification of the IHC KDM6B score regarding the EMT status. (**D**) The correlation between EZH2 and KDM6B scores in NSCLC. (**E**) The KM-plotter for *KDM6B* expression showing the survival curves of patients with lung cancer classified in high expression (red) or low expression (black) groups. (**F**) The KM-plotter for *EZH2* expression. **(G**) The KM-plotter for the mean of *EZH2*/*KDM6B* expression. ** = 0.005 < *p* < 0.01; *** = 0.001 < *p* < 0.005; **** = *p* < 0.001.

**Figure 2 cancers-12-03649-f002:**
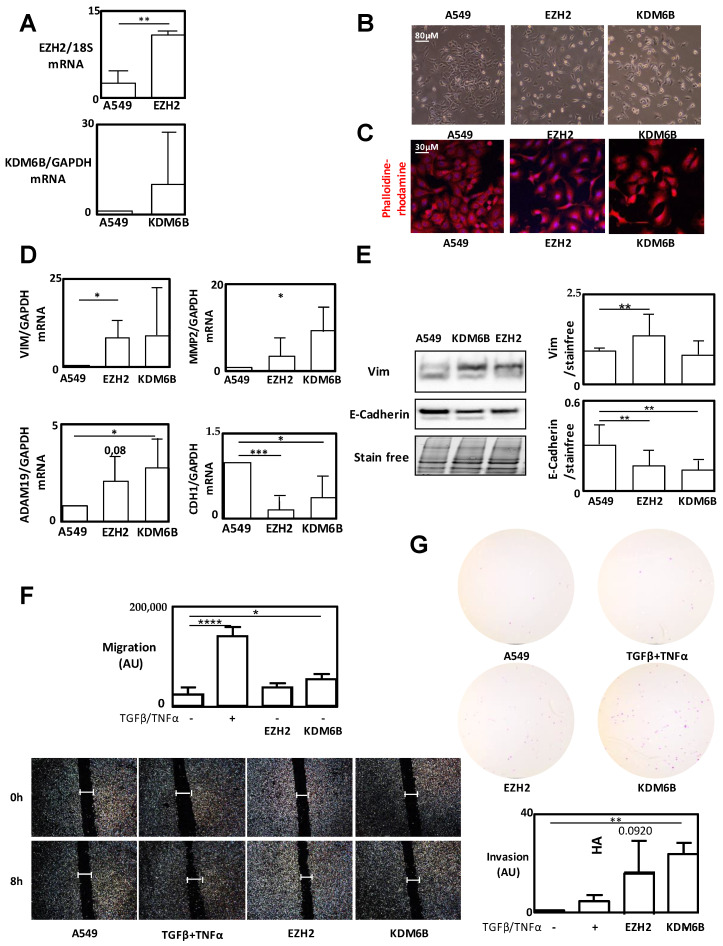
Overexpression of EZH2 and KDM6B induced EMT. (**A**) mRNA levels of *EZH2* and *KDM6B* after stable transfection of A549 with EZH2- or HA-KDM6B-expressing vectors. (**B**) Representative images showing the phenotype of A549 WT cells and A549 cells overexpressing EZH2 or KDM6B (EVOS™ 10× Objective). (**C**) Phalloidin-rhodamin staining of A549 WT or A549 cells overexpressing EZH2 or KDM6B. (**D**) Quantification of *Vimentin*, *CDH1*, *ADAM19*, and *MMP2* expression levels using qRT-PCR in A549 cells treated with TGFβ/TNFα or in cells overexpressing EZH2 or KDM6B. (**E**) Western-blot analysis showing a decrease in E-Cadherin expression in A549 cells overexpressing KDM6B. (**F**) Wound healing experiments in A549 cells treated with TGFβ/TNFα or in cells overexpressing EZH2 or KDM6B and the relative quantification of the effects observed (EVOS™ 4× Objective). (**G**) Invasion tests in A549 cells treated with TGFβ/TNFα or in cells overexpressing EZH2 or KDM6B (EVOS™ 4× Objective). * = 0.01 < *p* < 0.05; ** = 0.005 < *p* < 0.01; *** = 0.001 < *p* < 0.005; **** = *p* < 0.001.

**Figure 3 cancers-12-03649-f003:**
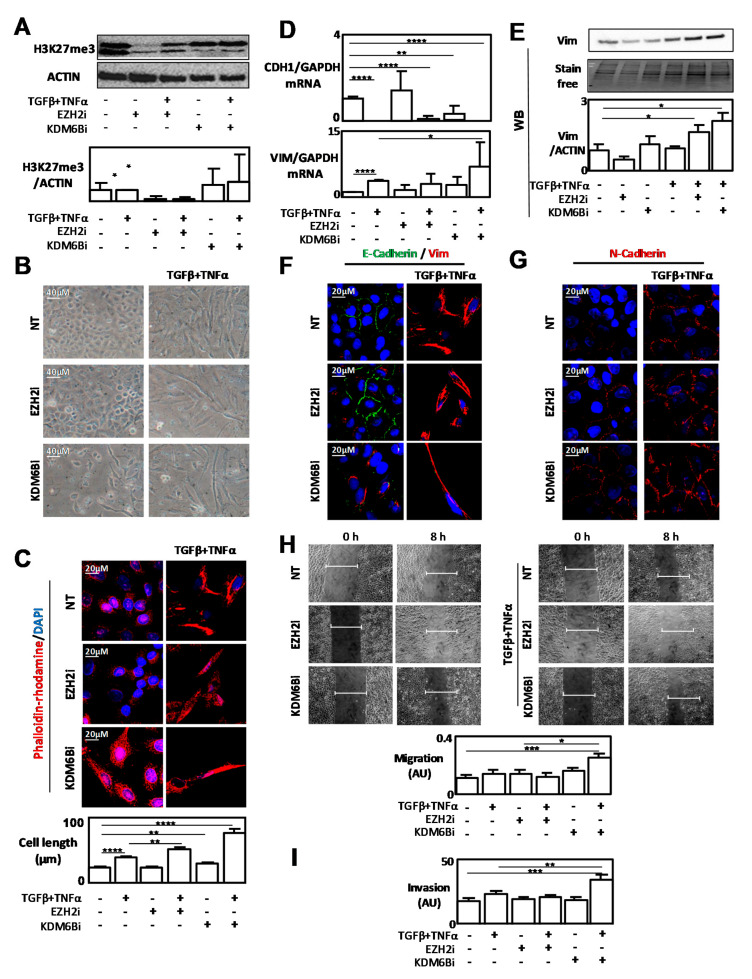
Inhibition of EZH2 and KDM6B activities using specific inhibitors favored EMT-like phenotypes. (**A**) Quantification of H3K27me3 levels using WB in A549 cells treated with TGFβ/TNFα and/or EZH2i. (**B**) Representative phenotype of A549 cells treated with TGFβ/TNFα and/or EZH2i or KDM6Bi (EVOS™ 10× Objective). (**C**) Phalloidin-rhodamin staining of A549 cells treated with TGFβ/TNFα and/or EZH2i or KDM6Bi. (**D**) Quantification of *CDH1* and *Vimentin* expression levels using RT-qPCR in A549 cells treated with TGFβ/TNFα and/or EZH2i or KDM6Bi. (**E**) Quantification of Vimentin expression using WB in A549 cells treated with TGFβ/TNFα and/or EZH2i or GSKJ4. (**F**) IF staining of E-Cadherin (green) and Vimentin (red) in A549 cells treated with TGFβ/TNFα and/or EZH2i or KDM6Bi. (**G**) IF staining of N-Cadherin (red) in A549 cells treated with TGFβ/TNFα and/or EZH2i or KDM6Bi. (**H**) Wound healing experiments in A549 cells treated with TGFβ/TNFα and/or EZH2i or KDM6Bi and relative quantifications (EVOS™ 4× Objective). (**I**) Invasion tests in A549 cells treated with TGFβ/TNFα and/or EZH2i or KDM6Bi. * = 0.01 < *p* < 0.05; ** = 0.005 < *p* < 0.01; *** = 0.001 < *p* < 0.005; **** = *p* < 0.001.

**Figure 4 cancers-12-03649-f004:**
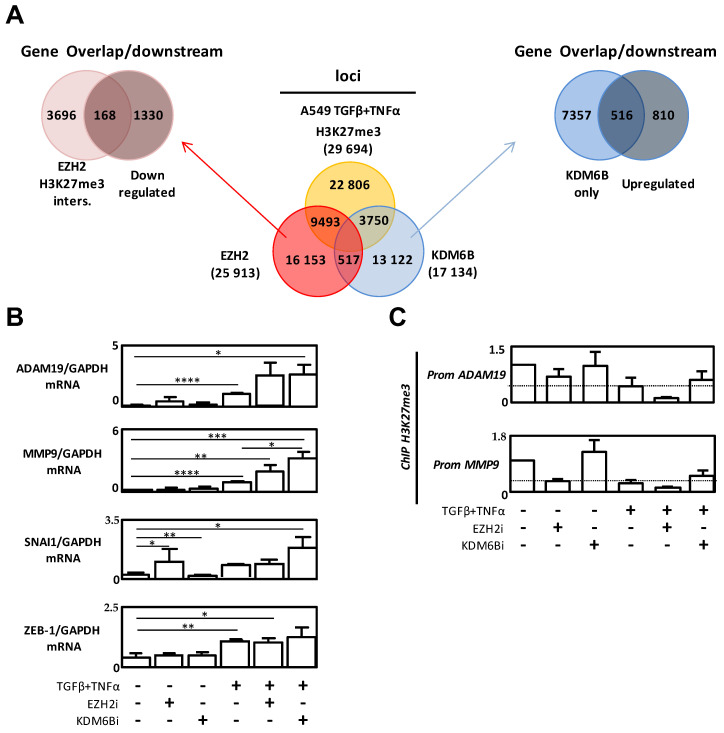
EZH2 and KDM6B regulated different genes during EMT. (**A**) Middle: Venn diagrams showing common and different DNA loci occupied by H3K27me3, EZH2, and KDM6B, and obtained following ChIP-seq experiments in A549 cells treated with TGFβ/TNFα. Left: A Venn diagram showing the intersection between genes co-occupied by H3K27me3 and EZH2 and down-regulated genes. Right: A Venn diagram showing the intersection between genes occupied by KDM6B alone and up-regulated genes. (**B**) Quantification of *ADAM19, MMP9, SNAI-1*, and *ZEB-1* expression levels using RT-qPCR in A549 cells treated with TGFβ/TNFα and/or EZH2i or KDM6Bi. (**C**) Quantification of the H3K27me3 content on *ADAM19*, *MMP9* using ChIP in A549 cells treated with TGFβ/TNFα and/or EZH2i or TGFβ/TNFα and/or KDM6Bi, respectively. * = 0.01 < *p* < 0.05; ** = 0.005 < *p* < 0.01; *** = 0.001 < *p* < 0.005; **** = *p* < 0.001.

**Figure 5 cancers-12-03649-f005:**
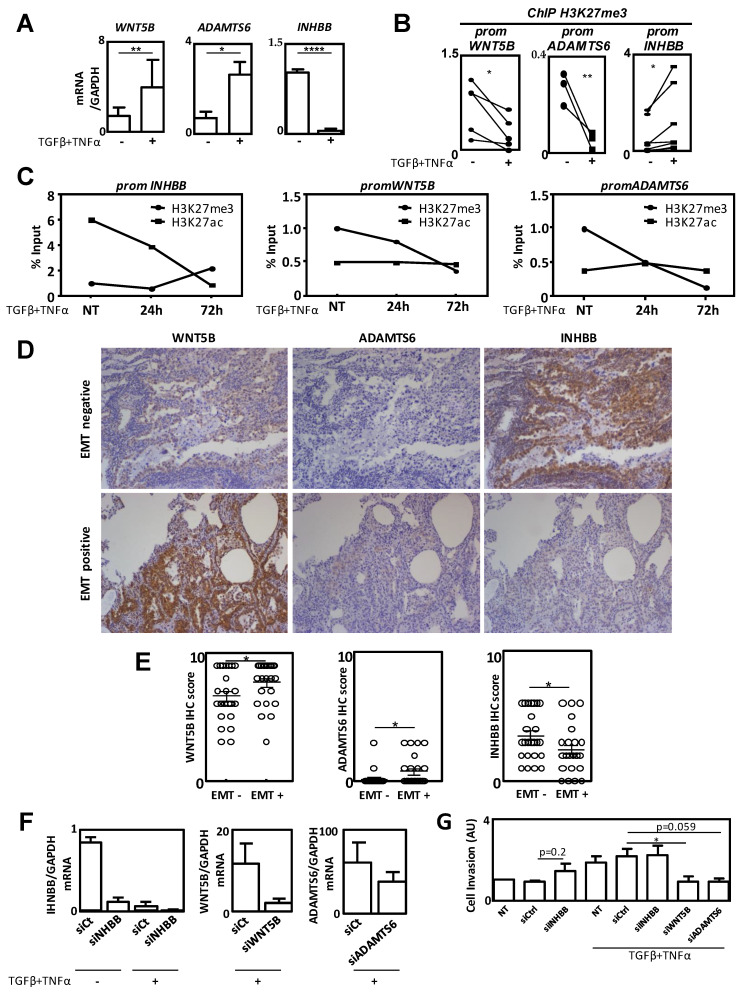
WNT5B, ADAMTS6, and INHBB were established as new epigenetic markers of EMT. (**A**) The validation of our biomarkers using classical ChIP targeting the H3K27me3 status on *WNT5B*, *ADAMTS6*, and *INHBB* promoters following TGFβ/TNFα treatment. (**B**) Validation using RT-qPCR of *WNT5B*, *ADAMTS6*, and *INHBB* expression following TGFβ/TNFα treatment. (**C**) Quantification of H3K27me3 and H3K27ac contents on *INHBB*, *WNT5B*, and *ADAMTS6* promoters following TGFβ/TNFα treatment for 24 to 72 h in A549 cells. (**D**) Representative IHC stainings of WNT5B, ADAMTS6, and INHBB in EMT-positive and EMT-negative NSCLCs (magnification × 100). (**E**) Quantification of IHC scores in regards to EMT status. (**F**) Validation of *siWNT5B*, *siADAMTS6*, and *siINHBB* using qRT-PCR. (**G**) Effects of *siWNT5B*, *siADAMTS6*, and *siINHBB* on A549 cell invasion. * = 0.01 < *p* < 0.05; ** = 0.005 < *p* < 0.01; **** = *p* < 0.001.

**Figure 6 cancers-12-03649-f006:**
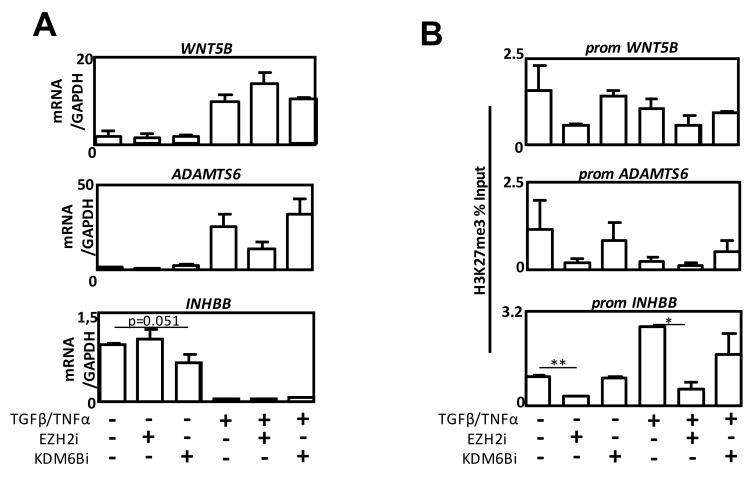
The effect of EZH2 and KDM6B inhibition on INHBB, WNT5B, or ADAMTS6 expression. (**A**) Quantification of *WNT5B, ADAMTS6*, or *INHBB* expression using RT-qPCR in A549 cells treated with TGFβ/TNFα and/or EZH2i or KDM6Bi. (**B**) Quantification of H3K27me3 content in *WNT5B*, *EZH2*, or *KDM6B* promoters using ChIP, in A549 cells treated with TGFβ/TNFα and/or EZH2i or TGFβ/TNFα and/or KDM6Bi. * = 0.01 < *p* < 0.05; ** = 0.005 < *p* < 0.01.

**Table 1 cancers-12-03649-t001:** A list of 40 most down-regulated genes in EMT associated with the H3K27me3 mark and EZH2 recruitment. Genes previously associated with EMT are underlined.

Overlap H3K27me3/EZH2 (ALIAS)	Full Name	Pathway	Downregulated FC
INHBB	Inhibin Beta B Subunit	Inflammation	−31.8
ANK3	Ankyrin 3	cytoskeleton	−25
CABLES1	Cdk5 And Abl Enzyme Substrate 1	cell cycle	−24
GREM2	Gremlin 2, DAN Family BMP Antagonist	differentiation	−12
VAV3	Vav Guanine Nucleotide Exchange Factor 3	cytoskeleton/signaling	−11.7
TC2N	Tandem C2 Domains, Nuclear		−11.5
HR	HR, Lysine Demethylase And Nuclear Receptor Corepressor	EPIGENETICS	−10.7
CPLX2	Complexin 2	vesicle Trafficking	−10.5
COBL	Cordon-Bleu WH2 Repeat Protein	cytoskeleton	−9.5
SLC5A11	Solute Carrier Family 5 Member 11	metabolism	−9.3
C9orf3	Chromosome 9 Open Reading Frame 3	M1 zinc aminopeptidase family	−8.4
CEACAM7	Carcinoembryonic Antigen Related Cell Adhesion Molecule 7	adhesion	−8.4
ABLIM1	Actin-Binding LIM Protein Family Member 1	cytoskeleton	−8
DBP	D-Box Binding PAR BZIP Transcription Factor	TF	−7.8
WNT6			−6.7
ANK1	Ankyrin 1	cytoskeleton	−6.5
EFCAB3	EF-Hand Calcium Binding Domain 3	calcium	−6.5
CXXC5	CXXC Finger Protein 5	TF	-6.1
FGFR3	Fibroblast Growth Factor Receptor 3	signaling	−6
MYO5B	Myosin VB	vesicle Trafficking	−6
CACNA1G	Calcium Voltage-Gated Channel Subunit Alpha1 G	calcium channel	−5.6
PMP22	Peripheral Myelin Protein 22	neurones	−5.4
PREX1	Phosphatidylinositol-3,4,5-Trisphosphate Dependent Rac Exchange Factor 1	signaling	−5.4
ANKRD33B	Ankyrin Repeat Domain 33B	cytoskeleton	−4.8
PAPLN	Papilin, Proteoglycan Like Sulfated Glycoprotein	peptidase	−4.6
C20orf196	Chromosome 20 Open Reading Frame 196		−4.5
NR5A2	Nuclear Receptor Subfamily 5 Group A Member 2	TF	−4.4
NRG2	Neuregulin 2	differentiation	−4.4
FOXA2	Forkhead Box A2	TF	−4.2
SYT12	Synaptotagmin 12	calcium/neurones/vesicle Trafficking	−4.2
LRP2	LDL Receptor Related Protein 2	Epithelial/signaling	−4.1
NTRK3	Neurotrophic Receptor Tyrosine Kinase 3	differentiation/signaling	−4.1
CXCL16	C-X-C Motif Chemokine Ligand 16	Chemokine	−3.9
EGLN3	Egl-9 Family Hypoxia Inducible Factor 3	hydroxylase	−3.9
KIAA1217			−3.9
CEBPA	CCAAT/Enhancer Binding Protein Alpha	TF	−3.8
HOXC13	Homeobox C13	TF	−3.8
NR2F2	Nuclear Receptor Subfamily 2 Group F Member 2	TF	−3.8
RASSF5	Ras Association Domain Family Member 5	signaling	−3.8
HRK	Harakiri, BCL2 Interacting Protein	apoptosis	−3.7

**Table 2 cancers-12-03649-t002:** A list of 40 most up-regulated genes in EMT and associated with KDM6B recruitment. Genes previously associated to EMT are underlined.

Overlap KDM6B (ALIAS)	Full Name	Pathway	Upregulated FC
LAMC2	Laminin Subunit Gamma 2	migration	136.0
MMP2	Matrix Metallopeptidase 2	invasion	84.0
IFI44	Interferon Induced Protein 44		85.0
PDPN	Podoplanin	adhesion/migration	70.2
EPHB1	EPH Receptor B1	signalling/diffenciation	61.1
ADAMTS6	ADAM Metallopeptidase with Thrombospondin Type 1 Motif 6	invasion	60.8
PLXNA2	Plexin A2	neurones/cytoskeleton	51.3
CORO2B	Coronin 2B	cytoskeleton	50.5
WNT5B	Wnt Family Member 5B	signaling	49.2
FGF1	Fibroblast Growth Factor 1	invasion	46.8
IL11	Interleukin 11	inflammation	40.9
SERPINE1	Serpin Family E Member 1	TGF signaling	36.2
SORBS2	Sorbin And SH3 Domain Containing 2	signaling	32.8
ST3GAL1	ST3 Beta-Galactoside Alpha-2,3-Sialyltransferase 1	golgi/metabolism	31.0
ADRB2	Adrenoceptor Beta 2	signalling	29.2
FLRT2	Fibronectin Leucine Rich Transmembrane Protein 2	migration/neurones	29.1
CCBE1	Collagen And Calcium Binding EGF Domains 1	invasion	27.7
ITGA1	Integrin Subunit Alpha 1	adhesion/inflammation	25.3
GABBR2	Gamma-Aminobutyric Acid Type B Receptor Subunit 2	signalling	25.1
CNTNAP2	Contactin Associated Protein Like 2	neurones/adhesion	24.9
TSPAN2	Tetraspanin 2	sgnaling/migration	23.5
MAMDC2	MAM Domain Containing 2		22.5
ABLIM3	Actin Binding LIM Protein Family Member 3	cytoskeleton	21.1
PDE11A	Phosphodiesterase 11A	signaling	22.0
SPOCK1	SPARC/Osteonectin, Cwcv And Kazal Like Domains Proteoglycan 1		18.5
FHOD3	Formin Homology 2 Domain Containing 3	cytoskeleton	17.9
LPAR5	Lysophosphatidic Acid Receptor 5	signalling	17.8
KIF26B	Kinesin Family Member 26B	vesicle trafficking/cancer	17.6
STC1	Stanniocalcin 1	metabolism	15.4
FILIP1L	Filamin A Interacting Protein 1 Like	cell division/invasion regulation	15.4
GPAM	Glycerol-3-Phosphate Acyltransferase, Mitochondrial	metabolism	15.3
FRMD6	FERM Domain Containing 6		14.9
COL1A1	Collagen Type I Alpha 1 Chain	invasion	14.1
PID1	Phosphotyrosine Interaction Domain Containing 1		14.0
NTN1	Netrin 1	migration	13.9
CDCP1	CUB Domain Containing Protein 1	invasion	13.3
SCG5	Secretogranin V	protein transport	12.4
TNS1	Tensin 1	adhesion	12.0
CUX2	Cut Like Homeobox 2	TF	11.9
COL13A1	Collagen Type XIII Alpha 1 Chain	invasion	11.9

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
