# Peer review of "EZH2 and KDM6B Expressions Are Associated with Specific Epigenetic Signatures during EMT in Non Small Cell Lung Carcinomas"

_cancers, 2020, doi:10.3390/cancers12123649_

Round 1
Reviewer 1 Report
The article by Lachat et al. is aimed at investigating the role of two enzymes, EZH2 and KDM6B, in EMT process. The study is of interest, however, it need some improvements in the research design and results presentation.
Major concerns:
- 2. section: it is not clear how the stable transfection of EZH2 (EZH2) or KDM6B (KDM6B-HA) was obtained, and which plasmid was used for EZH2 (maybe pMSCV-EZH2-PGK-Puro-438 IRES-GFP?). Usually stable transfection are carried out by the use of a selective drug, which enable the selection of the sole transfected cells. Please explain it properly in M&M.
- Figure 2.B and C: the images should report also the morphology and Phalloidin-rhodamin staining of untreated control cells (A549). By the legend, it appears that only A549 cells treated with TGFβ/TNFα are presented.
- Figure 2: the IF images (Fig 2.D,E) are not in line with the description in the text. The overexpression of both EZH2 or KDM6B by stable transfection does not emerge from the images, neither the increase of N-cadherin. Western blot analysis should be carried out to better quantify the actual expression of both enzymes and of all the selected markers, with respect to un-transfected cells and TGFβ/TNFα-treated cells.
- Figure 2.G. Representative images for the invasion assay should be added.
- Figure 2. The legend reports as B panel ‘the Phalloidin rhodamin staining of A 549 cells treated with TGFβ/TNFα or overexpressing KDM6Bm’. However, these data are not present into the figure. Please check the figure and the legend.
- The role of the KDM6Bm mutant is intriguing, suggesting that the expression, rather than the activity, could activate ETM. However, can the Author exclude that the transfection itself can activate such process? Do they perform experiments with empty vectors? This is never reported into the text.
- Section 2.3 The Authors emphasize the effect of both inhibitors on EMT process. However, nor the vimentin, the cell length, migration and invasion assays show significant results in comparison to untreated cells. The most relevant effects were obtained following TGFβ/TNFα treatment, alone or in combination with the inhibitors. The entire section (and the discussion) must be rewritten accordingly.
- Fig. 6. At my opinion, the western image representing Vimentin at panel E is a collage between two different experiments (is clearly visible a separating line between CRISP/EHZ2 and CRISP/KDM6B samples). Please provide a western blot image where the samples are loaded and probed at the same time.
- The CRISP models should be better characterized at molecular level. Expression of other EMT markers must be carried out.
- Fig. 7. The migration assay presents very poor quality images; it is not possible to distinguish populated from unpopulated areas of the scratch. Please provide better quality images.
Minor points:
- In the title, the NSCLCs acronym should be substituted by the complete form ‘Non Small Cell Lung Carcinomas’.
- Besides acronyms, the names of proteins (i.e. Vimentin, E-cadherin) should be written in lowercase letters. Please check throughout the manuscript.
- Line 72: the sentence ‘Similarly, although KDM6B is less studied, its….’ is misleading. It should be changed in: Similarly, KDM6B, although less studied, has been positively associated to EMT…
- Line 85: ‘were’ should be changed with ‘was’.
- Please check throughout the manuscript for grammar and type errors.
- M&M: the paragraph about Quantitative RT-PCR should be shifted after the invasion and migration assay.
Author Response
Added in attached file

Reviewer 2 Report
Lachat et al. report on the function of opposing epigenetic enzymes EZH2 and KDM6B. This is the first study of these proteins in the same system, which helps resolve some confusion about reports demonstrating opposing effect. The report is very thorough (6 figures with many panels each, 2 large tables, and another 8 supplementary figures each with many panels). The authors begin with observation of expression in human tumors, define expression in a cell model using A549 cells, target EZH2 and KDM6B with both overexpression and inhibition, define how each protein alters gene expression in the cell model, assess the function of specific target genes in EMT with overexrpession and inhibition experiments, and define the expression of these markers in human tumor samples. The amount of work is commendable and deserves to be published.
There are a few points that would improve this article. First is that the readability can be improved. Despite the added work, I suggest this only so that it does not detract from the effort put into this study. I would hate to see reduced dissemination of the results because of writing style.
Second, the discussion of the KDM6B point mutation findings should include another possibility besides that the enzymatic activity is unimportant. That is that some point mutants may have some kind of dominant negative effect in cells, which would induce EMT behavior in a manner similar to resulted yielded using inhibitor or knockdown approaches. Simply acknowledging this possibility, which is consistent with the findings throughout the paper, is enough.
Third, the conclusion that the roles of EZH2 and KDM6B are interconnected in the concluding sections is perhaps too strong. The fact that results have been obtained by targeting these proteins for overexpression or inhibition separately cuts against this conclusion somewhat. A better conclusion, especially in light of the authors very interesting ChIP-Seq experiments is to say that these two proteins work via distinct mechanisms that are likely complimentary.
Related to the third note, it would be interesting to see how combined overexpression of both proteins might affect EMT in this and other cell models, or how overexpression of one protein might be affected by inhibition of the other system. I mention this only to provide ideas to the authors, though these have undoubtedly been discussed, since suggesting such studies would clearly be beyond the scope of this publication.
Author Response
Added in enclosed file

Round 2
Reviewer 1 Report
The manuscript has been greatly improved, and it can be accepted in the present form